# Endocannabinoid System and Exogenous Cannabinoids in Depression and Anxiety: A Review

**DOI:** 10.3390/brainsci13020325

**Published:** 2023-02-14

**Authors:** Ahmed Hasbi, Bertha K. Madras, Susan R. George

**Affiliations:** 1Department of Pharmacology and Toxicology, Temerty Faculty of Medicine, University of Toronto, Toronto, ON M5S 1A8, Canada; 2McLean Hospital, Belmont, MA 02478, USA; 3Department of Psychiatry, Harvard Medical School, Boston, MA 02115, USA; 4Department of Medicine, Temerty Faculty of Medicine, University of Toronto, Toronto, ON M5S 1A8, Canada

**Keywords:** cannabinoid, endocannabinoid system, cannabinoid receptors, CB1R, depression, anxiety, cannabis, THC, CBD

## Abstract

**Background:** There is a growing liberalization of cannabis-based preparations for medical and recreational use. In multiple instances, anxiety and depression are cited as either a primary or a secondary reason for the use of cannabinoids. **Aim:** The purpose of this review is to explore the association between depression or anxiety and the dysregulation of the endogenous endocannabinoid system (ECS), as well as the use of phytocannabinoids and synthetic cannabinoids in the remediation of depression/anxiety symptoms. After a brief description of the constituents of cannabis, cannabinoid receptors and the endocannabinoid system, the most important evidence is presented for the involvement of cannabinoids in depression and anxiety both in human and from animal models of depression and anxiety. Finally, evidence is presented for the clinical use of cannabinoids to treat depression and anxiety. **Conclusions:** Although the common belief that cannabinoids, including cannabis, its main studied components—tetrahydrocannabinol (THC) and cannabidiol (CBD)—or other synthetic derivatives have been suggested to have a therapeutic role for certain mental health conditions, all recent systematic reviews that we report have concluded that the evidence that cannabinoids improve depressive and anxiety disorders is weak, of very-low-quality, and offers no guidance on the use of cannabinoids for mental health conditions within a regulatory framework. There is an urgent need for high-quality studies examining the effects of cannabinoids on mental disorders in general and depression/anxiety in particular, as well as the consequences of long-term use of these preparations due to possible risks such as addiction and even reversal of improvement.

## 1. Introduction

There is evidence that humans have used *Cannabis sativa* since ancient times for recreational and/or medicinal purposes [1]. Although it is still considered illicit by many, multiple countries around the world are softening their position by legalizing the use of these plants or their extracts for medicinal and recreational use. Historically, the identification and isolation of the main psychoactive ingredient of cannabis, Δ9-tetrahydrocannabinol (Δ9-THC) [2] led to the discovery of cannabinoid receptors and, subsequently, led to the identification of endogenous cannabinoids (eCBs), their synthesis, regulation, and the revelation that a complete endogenous endocannabinoid system (ECS) exists, as will be described. Medicinal usage was incorporated for a variety of ailments, ranging from appetite stimulation, insomnia, pain, vomiting, nausea and for other conditions such as cancer, post-traumatic stress disorder, anxiety and depression, to cite a few.

The purpose of this mini-review is to explore the association between depression or anxiety and the dysregulation of the ECS, and the use of phytocannabinoids and synthetic cannabinoids in the remediation of depression/anxiety symptoms. Depression is a widespread debilitating mental illness affecting approximately 300–350 million people globally [3]. Depression is often not an isolated disorder and may be co-morbid with other ailments such as anxiety, cancer, diabetes, stress, pain, and other neuropsychiatric disorders [4,5] for reviews). It is well known that other systems, such as dopamine, serotonin, opioid, adrenergic systems, are also involved in depression and anxiety, but the main focus of the present review is on the endogenous and exogenous cannabinoids in relation to depression and anxiety.

We describe the most important constituents of *Cannabis sativa*, review the endocannabinoid system, including the different cannabinoid receptors, and the machinery involved in the synthesis, transport and degradation of the endocannabinoids. This is followed by the most established evidence regarding depression/anxiety and ECS, before reviewing the relevant effects of different exogenous cannabinoid preparations in relation to depression and related ailments.

## 2. Constituents of *Cannabis sativa*

The chemical nature of the constituents of *Cannabis sativa* is diverse, with the number of natural compounds totaling 565 [1,6]. Among these, 120 cannabinoids have been isolated and classified into 11 general types, of which, the best known and the most studied are Δ9-trans-tetrahydrocannabinol (THC), and cannabidiol (CBD). THC, first reported by [2], is considered the principal psychoactive component, whereas CBD, first reported by [7], is recognized as a major non-psychotropic constituent of *Cannabis sativa* [8]. The discovery of these components was followed by the identification of their presumed targets through the cloning of specific cannabinoid receptors. Consequently, the endogenous ligands to these receptors were found in the brain, and called endocannabinoids (eCB), which triggered enormous progress in the exploration of their physiological importance, as well as the relevance of cannabis and its products in many health conditions including mental health [9,10,11].

## 3. Cannabinoid Receptors

Two receptors have been identified as the main cannabinoid receptors, the CB1 and CB2 receptors. The receptors have been cloned [12,13], established as the primary targets for endogenous and exogenous cannabinoids [10,14,15], and shown to belong to the superfamily of G-protein coupled receptors (GPCRs), characterized by seven transmembrane domains and coupled to G-proteins, notably to Gi/Go proteins [14]. In humans, the genes encoding the CB1 (cnr1) and CB2 (cnr2) receptors are located on the 6q14-q15 and 1p36.11 chromosomes, respectively [10]. The two receptors share ~44% sequence identity and diverge significantly in their distribution. CB1 receptor (CB1R) is highly expressed in different brain regions [16,17,18] and is considered the most abundant GPCR in the brain (reviewed in [10,14,15]). The CB2 receptor (CB2R) is principally expressed in immune tissues, inflammatory cells and in much lower densities in brain [10].

## 4. Additional Receptors for Cannabinoids

Beside their actions through the CB1 and CB2 receptors, the endocannabinoids, as well as the exogenous cannabinoids, bind also to other receptors [15]. Among these are the transient receptor potential vanilloid 1 (TRPV1) cation channel, peroxisome proliferator-activated receptors (PPARs, [19]), and at least three orphan GPCRs: GPR55, that we have cloned [20] and which has been reported as a cannabinoid receptor [21]; GPR18; and GPR119 [22,23,24]. It is also important to note that many receptor heteromeric complexes have been established in vitro and in vivo, as reviewed in [24,25], such as the CB1R–CB2R heteromers [26], and between CB1R or CB2R and a large variety of other receptors. Notable are the CB1R heteromers with dopamine D2 receptor [27,28], serotonin receptors 5HT1A [29] and 5HT2A [30], adenosine receptor A_2A_ [31], and GPR55 [32]. All of this diversification of signaling through the cannabinoid receptors and the range of heteromers may explain the varied involvement of the cannabinoid system in many pathophysiologic conditions and may be responsible for the variance in response to endogenous and exogenous agonists and pharmacological modulators of the cannabinoid system.

## 5. The Endocannabinoid System

The endocannabinoid system (ECS) consists of the cannabinoid receptors, the endogenous cannabinoids (eCBs), the enzymes involved in the synthesis and degradation of eCBs and the protein(s) involved in the reuptake of secreted eCBs.

### 5.1. Endocannabinoids

The primary eCBs are anandamide (AEA) and 2-arachidonoylglycerol (2-AG) (reviewed in [5]). Anandamide (*N*-arachidonoylethanolamine) was the first endogenous cannabinoid-like substance to be identified [33], which was then followed by 2-AG [34,35]. Both AEA and 2-AG were characterized as endogenous ligands by their capacity to bind and modulate CB1R and CB2R. Other endogenous molecules capable of modulating CB1/CB2 receptors have also been identified [36] and these include 2-AG ether [37], *N*-arachidonoyl-dopamine [38], O-arachidonoyl ethanolamine, [39], and oleamide [40]. However, the physiological roles of these latter substances have not been completely established yet [5].

### 5.2. Enzymes Responsible for the Synthesis and Degradation of eCBs

Endocannabinoids are lipophilic molecules synthesized ‘on demand’, meaning synthesis occurs after neuronal activation, from membrane phospholipids, and are released immediately, without storage in vesicles [5]. Both anandamide and 2-AG are produced by post-synaptic neurons (Figure 1A). Anandamide is produced in a two-step process and its levels are regulated by its degradation by the fatty acid amide hydrolase (FAAH) [41]. Also synthesized is 2-AG, in a two-step process involving the action of phospholipase C (PLC), followed by the hydrolytic activity of diacyl glycerol lipase (DAGL) [35]. The levels of 2-AG are essentially regulated by monoacylglycerol lipase (MAGL) [42]. There is a difference however in the location in which both eCBs are degraded. The hydrolysis of 2-AG occurs in presynaptic neurons by MAGL, after CB1R activation, whereas anandamide is hydrolyzed in postsynaptic neurons by FAAH, which may lead to a termination of anandamide action where it is synthesized [5]. In addition to hydrolysis, 2-AG can also be transformed into other bioactive compounds by cyclooxygenase-2 [43] and lipoxygenase [44]. Further, the metabolism of anandamide by lipoxygenase and cyclo-oxygenase enzymes generates oxygenated products that signal through non-cannabinoid targets [5,45].

## 6. Signaling of the ECS

The mode of action of the eCBs on their receptors (Figure 1) reveals that eCBs synthesized in postsynaptic neurons act pre-synaptically at CB1Rs in a retrograde mode [46]. There are data showing that eCBs can also activate localized CB1Rs post-synaptically [47,48]. In general, activation of CB1 receptors by eCBs decreases the probability of release of neurotransmitters through multiple mechanisms, comprising an inhibition of calcium influx and an activation of potassium channels [46,47,48,49]. These effects are terminated by eCB reuptake followed by degradation; 2AG by MAGL pre-synaptically and AEA by FAAH post-synaptically, as mentioned above. Furthermore, a large proportion (45–48%) of synapses in different brain regions are tripartite [47] (Figure 1B), formed by a pre-synaptic neuron, a post-synaptic neuron, and a glial cell, usually an astrocyte, which expresses CB1R [47]. In response to neuronal activation, eCBs are released by the postsynaptic neurons and activate CB1R expressed on astrocytes. The astrocytes respond with an increase in calcium from intracellular stores leading to the release of gliotransmitters (Figure 1B), capable of modulating the presynaptic and postsynaptic circuit elements [47,50,51]. Through these mechanisms eCBs can inhibit pre-synaptic neurotransmitter release at both GABA and glutamate terminals, thus modulating several neurotransmitter systems [5,47,52]. As illustrated for dopamine neurons, CB1 receptors expressed at presynaptic glutamatergic and GABAergic inputs to DA neurons may facilitate or suppress dopaminergic neuronal activity by modifying such inputs [53,54,55]. CB1 receptors on GABAergic terminals (Figure 2A) can facilitate dopaminergic activity through suppression of the inhibitory input onto GABA receptors present on DA neurons [56,57,58,59], leading to an increase of DA release in areas such as the nucleus accumbens (NAc) of the ventral striatum.

In contrast, CB1 receptors at glutamatergic synapses, suppress excitatory inputs onto the glutamatergic receptors on the DA neurons [57,60,61]. This mechanism involves glutamate release at synapses in prelimbic frontal cortex and in NAc at D1-enriched medium spiny neurons (D1-MSNs) (reviewed in [62]). Activation of the prefrontal glutamate afferents can lead to long-term depression (LTD) of NAc glutamatergic synapses, an effect also involving 2-AG release, presynaptic CB1-receptor and postsynaptic metabotropic glutamate receptor 5 (mGluR5) activation [62,63,64]. Mechanistically (Figure 2B), the activation of mGluR5 by released glutamate leads to the synthesis and release of 2-AG, which then retrogradely activates CB1 receptors to inhibit additional glutamate release (reviewed in [62]). This activation of CB1R by eCBs is usually accompanied by LTD of neurons [65].

## 7. Reuptake of eCBs

It has been suggested that eCBs are removed from the synaptic space by an uptake mechanism followed by hydrolysis [66], in a process similar to that which happens with other neurotransmitters or neuromodulators such as the well-known example of dopamine re-uptake by dopamine transporter (DAT). However, specific ‘endocannabinoid membrane transporters’, have not yet been identified [5].

## 8. Cannabinoids in Depression and Anxiety

Because depression and anxiety commonly accompany multiple other disorders, their etiology can be diverse and includes perturbations of multiple systems, in different brain regions; thus, it is difficult to identify the specific roles of cannabinoids in the induction or relief of depression and anxiety. In order to clarify this and simplify the analysis, we propose to approach this review of the role of cannabinoids in depression/anxiety in different sections as follows. First, preclinical data related to cannabinoids and depression- and anxiety-like effects, and stress in animal studies, followed by description of the perturbations of the ECS in humans with depression/anxiety. A third part will highlight the most relevant clinical observations obtained from long-time users of cannabis. Finally, a brief review of the clinical observations resulting from the use of cannabinoids to combat major depression disease (MDD), bipolar disease (BD), and depression/anxiety comorbid with other diseases.

### 8.1. Cannabinoids and Depression-like Effects in Animal Studies

In preclinical studies using animal model systems, mostly in rodents, cannabinoids, whether exogenous or endogenous, seem to generally have an antidepressant-like effect. However, opposite effects were occasionally observed, due essentially to differences in the dosage, the species, the strains and the experimental design considered. For more detailed reviews see [10,67].

There are a few animal tests used to investigate depression-like and anxiety-like behaviors, such as the forced swim test (FST) and the tail suspension test (TST), to measure manifestations of helplessness, despair and stress engendered by the two tests. The elevated plus maze test (EPM), evaluates the anxiety-like state of the animal subject. The light/dark box is another test that measures the anxiogenic/anxiolytic state of the subject. Another aspect tested is the hedonia/anhedonia level after treatments, which can be assessed using the sucrose preference test (SPT). All these tests can monitor the status at basal level, after administering a drug or after an experimental maneuver, such as after chronic unpredictable mild stress (CUS).

Several lines of evidence suggest that acute enhancement of CB1R activity results in antidepressant- and anxiolytic-like effects in rodents (Table 1). For example, activation of CB1R by administration of its agonists HU210 or oleamide resulted in antidepressant-like action in the rat FST model [68]. Similar antidepressant and/or anxiolytic effects were observed after acute activation of CB1R by THC [69,70] or other cannabinoids, such as the synthetic agonists WIN55, 212–2 [71], CP55, 940 [72], as well as the endogenous AEA [73,74]. Other cannabinoid agonists or partial agonists, endogenous and exogenous, also showed similar effects (reviewed in [10,67]). The efficacy of these cannabinoid compounds was as good as some known antidepressants with which they were compared. In most cases, these positive effects on anxiety and depression were observed at relatively low doses of cannabinoids activating CB1R, whereas higher doses were often pro-depressive and anxiogenic [69,75,76,77,78,79]. This observation could mirror the biphasic effects usually seen in humans [10,69]. There are, however, some reported differences, which were essentially explained by species (mouse vs. rat) and strain differences, the brain region injected or studied, the test used, or other experimental differences [10,67].

### 8.2. Evidence for the Involvement of the ECS in Depression, Anxiety, and Stress

The involvement of the CB1R in the above-described effects of cannabinoids was usually verified by blockade of the receptor, using an antagonist/inverse agonist, such as the most known rimonabant [86] or AM251 [86]. However, opposing results have also been shown by some studies in which it was reported that the antagonist SR141716A (structurally comparable to rimonabant) and AM251 exerted antidepressant-like effects in animal models [87,88,89,90]. In the case of rimonabant, the effects were similar to those of antidepressant fluoxetine in various animal models of depression [90]. Moreover, such an effect was absent from the gene-deleted CB1−/− mouse treated with AM251, indicating a role for the CB1 receptor in depression. The mechanism underlying the antidepressant-like effects of rimonabant and other antagonists or inverse agonists remains to be determined, especially since rimonabant in humans has shown pro-depressive, pro-anxiogenic and even ideations of suicide, as will be described later. In contrast, multiple animal studies have shown that the blockade of CB1R signaling in rodents results in anhedonia-like behavior [91], and heightened levels of basal anxiety [92,93,94], and in general, an increased tendency to develop a passive coping response to stress [95,96]. In these latter studies, an increase in the hypothalamic–pituitary-adrenal axis (HPA axis) activity under both basal conditions and following exposure to stress was also observed [86,95,96,97].

Models of depression and anxiety involving stress have shown that the function of the ECS is reduced. For example, CB1R has been shown to be decreased in several brain structures implicated in depression in rats exposed to chronic unpredictable stress (CUS) [98,99,100]. When tested, the levels of eCBs, notably AEA, were reduced in different brain structures in studies of the CUS model of depression [97,98,99,100,101,102], demonstrating that chronic stress paradigms result invariably in a loss of AEA signaling. In addition, chronic stress exposure dramatically decreased the aptitude of CB1R to inhibit neurotransmitter release in regions involved in depression, such as the striatum and hypothalamus [103,104,105]. Additional evidence for the role of the ECS, and precisely the CB1R, in depression in preclinical studies came from knock-out mouse models in which CB1R (CB1R-KO) was eliminated. To simplify, studies from this model showed that decrease or elimination of CB1R or its signaling resembles the effects of antagonists, in that it is pro-depressive and pro-anxiogenic overall.

Other evidence derives from the use of a fatty acid amide hydrolase (FAAH) inhibitor or inhibitor of reuptake of eCBs. Chronic administration of the FAAH inhibitor URB597, which would increase AEA levels, has been shown to reduce anxiety-like effects and to display antidepressant-like actions in a CUS rat model; effects that were prevented by a CB1R-selective antagonist SR141761A [106]. This effect was not always observed [107], as URB597, an irreversible FAAH inhibitor, failed to demonstrate antidepressant-like actions in either the FST or TST. The use of other FAAH inhibitors or the use of FAAH gene deletion (FAAH-KO) model mouse, however, showed clearly that inhibition (or ablation) of FAAH was anxiolytic and anti-depressive (reviewed in [68]). These results are corroborated by the use of an inhibitor for eCB uptake, AM404. Accumulation of eCBs following administration of the uptake inhibitor AM404 elicited an antidepressant-like effect, presumably by indirect stimulation of the CB1 receptors by eCBs [68].

Thus, most preclinical studies underline an important role for the ECS in anxiety- and depression-like behaviors, and that the ECS seems to be compromised in the animal models of depression, anxiety, and stress exposure.

### 8.3. Perturbations of the ECS in Humans with Depression

In humans, there are multiple lines of evidence suggesting a role for the ECS in depression, anxiety, and stress. Most of the available evidence for perturbations affecting ECS in depression and anxiety derive from two major sources: post-mortem tissues and clinical measurements of eCB levels, notably AEA and 2AG, in the blood or cerebrospinal fluid (CSF) of patients. The other observations are related to cannabis use for recreational or self-medication purposes, which will be treated independently.

Information pertaining to the role of ECS in human depression, anxiety and stress is relatively scarce, and not completely validated, as reviewed in [67,108,109]. From post-mortem studies, an upregulation of CB1R and its G-protein signaling has been observed in the dorsolateral prefrontal cortex of depressed suicide victims [110]. Elevated levels of the eCBs AEA and 2-AG, and CB1R-mediated G-protein signaling in the prefrontal cortex of alcoholic suicide victims were also observed [111]. The up-regulation of CB1R coupling to G-proteins, specifically to Gαo, and not Gαi or Gαz protein subunits, has also been observed in the prefrontal cortex of depressed suicide victims [112]. It has been revealed, however, that this up-regulated CB1R/Gαo functionality might be due in part to the administration of antidepressants [67,112]. An interesting point is that the fluctuations of the eCBs seem to be linked to the gravity of depression [67]. Increases in serum levels of 2-AG and/or AEA has been observed in patients with minor depression, whereas patients with major depression showed reduced serum levels of 2-AG and/or AEA. Similarly, CB1R levels seem to depend on the severity of the depression. For example, CB1R-immunopositive glial cells in the grey matter were decreased in major depression, whereas no evidence of an altered density of CB1R immunopositive cells in schizophrenia or bipolar disorder was found [113]. The evaluation of the circulating levels of eCBs in body fluids may give an indication as to the role of the ECS in these disorders, although, as mentioned above, the severity of the disease and pre-treatment with other drugs, including antidepressants, may play a role in the final levels quantified. For example, it has been reported that plasma 2-AG levels, as well as CB1 and CB2 mRNA levels, are elevated in the lymphocytes of osteoarthritic patients, with a positive correlation between 2-AG levels, pain and depression [5]. In a recent systematic review of different studies tackling this matter [108], it was reported that differential effects of stress, depression, and anxiety symptoms may modulate endogenous levels of the eCBs. For example, acute psychological stress increased AEA and other fatty acid ethanolamides in healthy volunteers [114], whereas, in a study involving PTSD patients [115] there was a hypoactive eCB system, notably with decreased AEA and 2-AG, among others, in the plasma of the participants after physical and psychosocial stress. In another study involving patients with treatment-resistant depression or bipolar disorder, before and after electroconvulsive treatment [116], it was noted that the treatment increased AEA in the CSF of patients. In a cohort of students who were carriers of the A allele of the FAAH enzyme, characterized by increased basal AEA, this elevation was shown to facilitate fear extinction and ameliorate extinction of recall [117]. Moreover, it was suggested that the polymorphism in the FAAH gene was associated with bipolar disorder and major depression [118] which is in line with the importance of FAAH activity in depression and suicidal behavior [111].

Reducing endocannabinoid signaling in humans is usually sufficient to produce depressive symptoms [119,120]. As an example, the CB1 receptor antagonist/inverse agonist, rimonabant, has been used as a treatment for obesity in humans. However, its administration led to severe symptoms of anxiety and depression in a significant proportion of individuals [121,122], which resulted in the removal of rimonabant from the market [119]. Another line of evidence comes from genetic studies that showed that some genetic polymorphisms in CB1R and CB2R may be associated with major depression and bipolar disorder, as reviewed in [119]. Some of these genetic studies also showed that single nucleotide polymorphisms (SNPs) in the CB1R can increase the vulnerability to the development of a depressive episode following exposure to life stress [123], and confer an increased risk of antidepressant resistance [124]. It has also been found that SNPs in the CB1 receptor gene can result in blunted neuronal activation in response to rewarding stimuli [124]. A significant increase in the frequency of these CB1 receptor SNPs in patients with mood disorders was also observed [118].

From these studies, it is clear that perturbations of the ECS play a role in depression, anxiety and response to stress. It also indicates that, in general, increasing eCB levels would be beneficial in some neuropsychiatric disorders, including MDD and probably chronic anxiety. This suggests also that blocking the degradation of the eCBs and/or activating the ECS may ameliorate these particular conditions.

## 9. Clinical Observations Obtained from Long-Time Cannabis Users

Recreational and potential medicinal usage of cannabis and derivatives is based on their perceived acute properties of inducing euphoria, relaxation and analgesia, as well as promoting sleep, appetite stimulation and other effects such as relief of chronic pain, nausea and vomiting. However, multiple adverse effects have been described in long-term users of cannabis, including possible development of psychosis [125] and increased depression and anxiety [108,126,127,128]. Evidence suggests that both short- and long-term use of cannabis heighten the risks for psychiatric illness. A clear example is the usual cognitive impairment observed in both acute and chronic cannabis users, which includes impaired short-term memory, motor coordination and control, sleep, executive functioning, as well as altered judgement, which, in sum, affect most aspects of regular day-to-day life of the users [125,129,130,131,132]. Additionally, high doses of cannabis, and more importantly, high content THC in some engineered cannabis plants and preparations can induce paranoia and psychosis even after single doses or short-term cannabis use and may even induce these in the long-term, structural brain modifications in younger brains [133,134]. Furthermore, chronic cannabis use has been associated with a higher risk of developing cannabis use disorder (CUD) and may lead to other substance use disorders (SUDs) [130,135,136,137], including heightened functional impairment of normal daily activities comparable to the harms observed with any SUD, as reviewed in [135,138,139]. A large proportion of long-term users would transition to CUD and most experience strong withdrawal symptoms after cessation [128,140,141,142,143], including irritability, anger or aggression, nervousness or anxiety, sleep disruption, decreased appetite, restlessness, or depressed mood, all of which could promote further drug seeking and relapse. Clinically, 50–95% of heavy users may suffer from some symptoms of cannabis withdrawal [140,141,142], with a meta-analysis reporting an overall prevalence of 47% [143]. Moreover, it has also been shown that long-term use of cannabis may lead to the use of other illicit drugs, which is reinforced by increased frequency and earlier onset of use [144], thus increasing the risk of SUDs and closely aligned mental illnesses, notably depression and anxiety. In a paradoxical twist, the use of cannabis to self-medicate for depression and anxiety may in fact lead to CUD (or SUD), with individuals suffering with a mental illness generating higher dependence on cannabis as a coping strategy for the management of these symptoms and consequently augmenting the risk of developing CUD (or SUD), which in turn may lead to an augmented risk of exacerbating severe depression and anxiety. These effects can be worsened by different factors, such as an earlier age of onset of use, the quantity and potency, the frequency, as well as the period of use and percentage of THC in the preparation [11,145]. In any case, these effects are essentially due to the actions of THC acting as a partial agonist at CB1R [146,147]. In contrast with THC, the use of CBD, the second most abundant cannabinoid in cannabis, has been shown to be devoid of any direct psychoactive behavioral effect in animal studies and, notably, had no rewarding effect, thus negating its potential for abuse as an addictive substance [148,149]. It has been shown that CBD can counter many of the actions of THC and may even provide a certain degree of protection against some, but not all, psychoactive and other deleterious properties of THC [11,145].

In relation to depression and anxiety, a higher incidence of depression and anxiety is usually the norm among long-term users of cannabis, although this could be due to other confounding and comorbid factors. For example, as reviewed in [128], in major depressive disorder (MDD) and bipolar disorder (BD), evidence generally reveals a harmful outcome of long-term cannabis use. Cannabis use was correlated with a higher risk of MDD diagnosis, and earlier onset of cannabis use was correlated with shorter time to MDD diagnosis [128,150]. This positive correlation between long-term cannabis use and depression or anxiety seems to be a consistent finding, although some other studies indicate this correlation disappears when controlling for multiple confounding factors, such as the co-use of other drugs, sex, age, genetic traits and education level, among others [128,151,152,153]. Similarly, evidence indicates the harmful effects of long-term cannabis use in bipolar disorder (BD), with frequent cannabis use being associated with heightened risk of BD onset [154] and/or exacerbation of BD symptomology [154,155,156]. It is worth noting that these effects of long-time use of cannabis are in opposition to the effects of acute cannabis use, which seems to lead to an improvement in BD symptoms [157]. As for MDD, there are studies that found no statistically significant correlation between cannabis use and BD consequences when other factors were considered [157,158]. Anxiety symptoms seem to be higher in cannabis users compared with non-users [159] and are enhanced with the earlier onset of cannabis use [159,160,161,162]. Another condition, which can be related to anxiety and depression, is post-traumatic stress disorder (PTSD), for which cannabis is being used either as self-medication or doctor-prescribed medication. While those undergoing acute or short time treatment, mostly medically assisted, seem to show some benefits in lowering anxiety levels and ameliorating the quality of life, it has been shown that daily cannabis use is positively associated with increased severity of PTSD symptoms [163]. Most of the evidence supports an association of long-time cannabis use with a higher risk of PTSD symptoms and heightened negative affect [128,163,164].

In sum, the acute use of cannabis may offer relief from various symptoms linked to anxiety, stress and depression. However, the long-term consequences of cannabis use seem to have higher potential for harmful outcomes for these conditions. Nevertheless, other confounding factors may play a role in heightening these risks. Indeed, as recently reported [109,164], the World Health Organization (WHO, [3]) and the National Academies of Science, Engineering, and Medicine [165] concluded that the effects of cannabis use on depression seem to be limited and slightly increase the risk of developing depression. Nevertheless, a systematic review [166] that appraised 44 previous systematic reviews, including 1053 different studies covering a broad spectrum of negative health outcomes directly connected to cannabis use, concluded that a clear association exists between cannabis use and psychosis, affective disorders, anxiety, sleep disorders, cognitive failures, respiratory adverse events, cancer, cardiovascular outcomes, and gastrointestinal disorders. Several studies have concluded that cannabis use during adolescence had a strong impact on risk for psychosis and incidence of schizophrenia as well as a causal association with poor school performance, lower educational attainment and early school dropouts, as reviewed in [166]. Many studies have also reported that cannabis use increased the risk for developing MDD and BD, as well as for maintaining high levels of anxiety over time, as reviewed in [128,166]. All these diverse harms that ensue from cannabis use have led different studies to conclude that cannabis use and misuse constitute a highly relevant public health challenge [166].

## 10. Clinical Use of Cannabinoids to Treat Depression, Anxiety, and Stress

Although the long-term effects of cannabis and its derivatives are not completely known, except on the part of the long-time users of cannabis themselves, the medical potential for cannabis and cannabis-based products have been suggested by multiple observational, open-label studies, and a few RCTs [108,128] (Table 2). The presumed medicinal indications cover a large array of conditions, including chronic pain, epilepsy, insomnia, cancer-associated pain, vomiting and nausea in cancer and HIV patients, PTSD, Parkinson’s disease (PD), depression, anxiety, and stress, among others [164]. These studies, however, contain multiple weaknesses, as several recent systematic reviews and meta-analyses have found [108,109,128]. Nevertheless, among the multiple constituents of cannabis, THC and CBD have been the most subject to different investigations in animal models and trials in humans. There are also some studies of synthetic cannabinoids or endogenous cannabinoids [5,108,128,167,168,169]. In general, cannabis and its derivatives seem to be well tolerated by humans and animals alike when used at relatively low doses and during short time periods [168,169,170]. Under these conditions, they are usually linked to many positive therapeutic effects for a wide range of symptoms and diseases [164,170]. Briefly described are the therapeutic effects reported for THC, CBD, and some synthetic cannabinoids and, for each, the available data related to effects on anxiety, stress and depression are presented.

### 10.1. Δ9-Tetrahydrocannabinol

The biphasic effect reported in animal studies, with low doses reducing anxiety and high doses producing anxiogenic-like and antidepressant-like effects [69,70,71,72] are also seen in humans. THC has demonstrated biphasic, dose-dependent effects on anxiety in healthy adults [173,174,175,176,177,178]. Neuroimaging data have also confirmed that THC can both increase [175] and decrease [174] emotional arousal/processing of negative stimuli at specific doses [178]. Certain THC doses globally induced anxiogenic effects in healthy individuals, although in this study half of the subjects were recreational users of cannabis [108,179]. However, due to the credible harmful effects over the long-term, such as addictive potential and the induction of anxiety and depression, multiple meta-analyses and reviews warn against the use of high doses of pure THC or cannabis-based preparations containing THC for long periods of time [108,128].

Other therapeutic effects of THC reveal stimulation of appetite, and it is used for this effect to treat patients with anorexia and cachexia of advanced cancer [180]. THC is also used in chemotherapy patients due to its effect of reducing nausea and vomiting [181,182]. THC has been shown to reduce acute and chronic pain (for a review see [5,170,183], including neuropathic pain [184,185,186]. THC has also been shown to help overcome some negative symptoms linked to PTSD in war veterans [167,187,188]. There are two synthetic analogs of THC approved by the US Food and Drug Administration (FDA) that could be prescribed for chemotherapy-induced nausea and vomiting: nabilone and dronabinol (reviewed in [189]). Nabilone has been shown to be effective in reducing vomiting and nausea [190], whereas dronabinol is currently being assessed for its analgesic properties in patients with breast cancer and in patients with chronic pain [5,167,189], anorexia of HIV/AIDS and cancer patients [191,192]. The antiemetic effects, consisting of the easing of nausea and/or vomiting have also been observed in animal models [189,193,194,195]. Amelioration of activity and motor impairment have been observed in animal models of Parkinson’s Disease (PD) [196] and Amyotrophic Lateral Sclerosis (ALS) [197].

### 10.2. Cannabidiol

Multiple positive effects have been linked to CBD (for recent reviews, see [167,189,198]. These include neuroprotective properties and reported effects as an anxiolytic and antipsychotic agent. Other potential therapeutic benefits attached to CBD include amelioration of symptoms in disorders such as PTSD, bipolar disorder, epilepsy, social phobia, schizophrenia, Parkinson’s disease and more [189]. Preclinical animal models suggest that CBD has beneficial effects by itself as well as reducing or abolishing some negative THC-induced effects, notably at long-term [108,128]. It also shows some benefit in reducing nausea and vomiting [199] and has an anticonvulsant effect, which is a promising potential for use, especially for children with epileptic syndromes [189,200].

Unlike THC, CBD has no psychoactive effect, but can in fact counter the psychoactive effects of THC and other negative effects, such as inhibiting the THC-induced cognitive impairment in preclinical studies involving non-human primates [201]. In depression, anxiety, and stress, CBD has constantly been shown to induce anxiolytic and antidepressive potential [108,128,178]. CBD has also been shown to counter many of the effects of THC, including THC-induced anxiogenic effects [173,174,175,176,178,202]. Healthy participants have also reported decreased levels of subjective anxiety 90-min post administration of CBD. Some anxiolytic effects of CBD have been seen to be similar to the post-stress anxiolytic effects of isapirone, a selective 5-HT1A receptor partial agonist known for its antidepressant and anxiolytic effects [178,202]. The positive effects of CBD have also been shown in multiple other studies, such as those with healthy volunteers under psychological stress [108,203]. Neuroimaging studies have also confirmed that CBD administration decreased activity in limbic and paralimbic regions during emotional face processing tasks [175,178,204]. In Parkinson’s patients, CBD ameliorated the motor symptoms and also improved the quality of life of these patients [205]. CBD also decreases anxiety in socially anxious individuals [206,207]. In sum, based on the accumulated data so far, CBD seems to have great therapeutic potential without the significant adverse psychoactive effects associated with THC and this is likely due to its multiple mechanisms of action. Of note, the precursor of CBD, Cannabidiolic acid (CBDA) seems to be 100–1000 times more potent than CBD in reducing toxin-induced vomiting and nausea in animal models [208,209] and has also been shown to act as an anxiolytic agent by preventing stress-induced anxiogenic-like behavior [210].

### 10.3. Combinations of THC and CBD

Combinations of THC and CBD have also been tested for different illnesses, and the results indicate that preparations with lower THC and higher CBD ratios are usually more efficient than the inverse (higher THC and lower CBD ratios) [11,145]. For example, Nabiximols, a cannabis-based preparation containing THC and CBD in a ratio of 1:1, was approved in Canada for the relief of Multiple Sclerosis (MS) spasticity or cancer pain [168,169]. Interestingly, in preclinical tests ineffectively low doses of THC or CBDA when given alone turn into very effective treatment for acute nausea and vomiting when given in combination [168]. These very low doses of combined CBDA and THC also showed anti-inflammatory effects and reduced pain sensation in these preclinical studies, although CBDA by itself was also capable of such effects [168].

### 10.4. Synthetic Cannabinoids

Human studies using synthetic cannabinoids are rare in the context of patients with depression and anxiety [108,128]. Depression and anxiety in most of these studies were evaluated only as secondary analyses within healthy populations or populations suffering from another illness such as pain, cancer, and PTSD [5,108,128,178]. Nabilone (Cesamet) and dronabinol, synthetic analogues of THC, are two FDA cannabinoids approved and used to treat different ailments, including nausea and vomiting, associated with cancer chemotherapy and the treatment of anorexia associated with weight loss in patients with AIDS as well as multiple sclerosis (MS) in different countries [108]. In relation to depression and anxiety, nabilone improved the quality of life and reduced pain in a randomized, double-blind, placebo-controlled trial in patients with fibromyalgia [211], and reduced anxiety and pain in cancer patients when used as an adjunct to opioid treatment [212]. These data are in line with results obtained in studies using dronabinol, another synthetic THC, in patients with chronic central neuropathic pain or fibromyalgia [5,213,214]. Dronabinol has been studied in healthy populations in a small double-blind randomized controlled trial (RCT) (*n* = 16; 7.5 mg) which showed reduced limbic reactivity to angry or fearful faces during an emotional faces processing task [178,215]. Dronabinol was also efficient in fear extinction, a key component of PTSD and anxiety disorders [216,217,218].

Few research studies involving clinically anxious/depressive populations were analyzed [108], but it was concluded that nabilone significantly reduced anxiety [219,220] as it was seen to be associated with a 26.5% improvement in “generalized anxiety scores” [220]. This small study, however, involved patients also taking an antidepressant medication for mixed anxiety and mood disorders [178].

## 11. Discussion, Conclusions and Perspectives

Compelling evidence indicates an important role for the endocannabinoid system (ECS) in the regulation of mood disorders, including anxiety and depression. The involvement of the ECS in mood disorders is linked to its wide presence throughout different brain structures involved in the regulation of mood, emotion, and reward. In addition, there are strong scientific indications showing a wide range of interactions between the ECS and many other endogenous transmitter systems, with its capacity to modulate GABAergic and glutamatergic tones, as well as the release of dopamine, serotonin, opioids, norepinephrine, among many other effects, including regulating astrocyte and glial functions. Changes affecting ECS signaling produce a plethora of physiological and behavioral effects consistent with symptoms of depression and anxiety, in line with the hypothesis that a deficit in endocannabinoid signaling may yield a susceptibility to, or directly participate in, the development of a depressive episode [169]. Perturbations affecting the levels of eCBs seem to affect depression and anxiety symptomatology in humans and associated behaviors in animal models. In severe mood disorders, such as MDD and BD, decreases in eCBs and/or cannabinoid receptor signaling have been shown, whereas increasing the endocannabinoid tone seems to be beneficial in reducing depressive and anxiogenic behavior and symptoms. This has been shown by different approaches, including by preventing eCB degradation by inhibitors for FAAH and preventing eCB action by antagonism of CB1R. One clear example, mentioned above, is the distinct insight provided from CB1R blockade using rimonabant as an anti-obesity drug, which necessitated its removal from the market because of induction of severe symptoms of depression and suicidality. In addition, SNPs for two major players in the ECS, FAAH and CB1R, were linked to depression and anxiety.

Nevertheless, the field is still unclear when it comes to recommendations regarding the use of cannabinoids in treating depression, anxiety, and effects of stress, whether as antidepressants or as adjuncts to other antidepressants. Cannabis and its derivatives from both plant-based and synthetic origins are used in many circumstances to treat pain, nausea/vomiting, stress, anxiety and depression co-morbid with serious diseases such as cancer, MS, PTSD, Parkinson’s disease and other illnesses. Although showing a positive effect of cannabinoids in ameliorating the quality of life of many patients, including an improvement in their mood in general, there are many weaknesses and flaws surrounding the studies conducted so far. Different systematic and meta-analysis reviews agree on these aspects and list among the flaws detected: the small sample size of the studies; the lack of RCTs; the use of diverse preparations; different dosages and methods of administration; and the use of healthy individuals, with no regard for sex and age differences. In some circumstances, these weaknesses have led to conflicting and opposing conclusions [102,122]. This is greatly magnified in the contemplation of the use of cannabis or THC-enriched preparations for long-term medicinal use. The long-term use of cannabinoids, notably cannabis or THC-rich products, has significant deleterious effects, especially among humans with addictive liability patterns and can cause serious and permanent harm in adolescents. As mentioned above, between 50% and 95% of heavy users show cannabis withdrawal symptoms [140,141,142,143], with a recent meta-analysis reporting an overall prevalence of 47% among such users [144]. Chronic or heavy cannabis use has been shown to lead to cannabis use disorder (CUD), with strong links to other substance use disorders (SUD) and the symptoms of anhedonia and anxiety that are usually attached to them. In MDD and BD, the conclusions from the meta-analyses were clearly negative regarding the use of cannabis or THC-preparations over the long-term [108]. Although CBD is present in some of these preparations and may inhibit some side effects of THC, there is no certainty that CBD can abolish them all. Furthermore, during the last two decades, the potency of cannabis products has escalated together with a tendency to increase THC and decrease CBD concentrations [11,133,221]. This fact will certainly amplify the negative impact of cannabis self-medication, especially among the young population and patients already severely impacted by depression, anxiety and stress. Some authors of systematic reviews and meta-analyses have concluded that there are no clear benefits in using cannabis and certain cannabinoid preparations in combatting depression and anxiety in the long-term [108,109,128]. This has been accompanied by warnings that long-term cannabis or preparations high in THC may in fact worsen these disorders [108,128]. There is a definite need for more RCT studies, especially in severe mood disorders such as MDD and BD, where there is a lack of such studies. There is expectation of some progress with future studies that consider these pitfalls. Some of these studies are ongoing or at the stage of recruiting subjects [108]. In the interim, it is very difficult to draw a definitive conclusion in relation to the use of cannabinoids for treatment of depression and anxiety. Nevertheless, a common conclusion is that short-term judicious use of cannabis or derivatives lower in THC and higher in CBD may be beneficial in reducing anxiety and depression in different disorders. However, the long-term prospect of using such cannabis-based medicinal preparations may need much more careful scrutiny as it may result in the converse of the desired effect. In addition, the possibility of inducing cannabis use disorder or addiction to cannabis and cannabinoid preparations rich in THC is a distinct possibility that should not be overlooked among certain patients, especially since the prevalence of such disorders is high and rising [144]. CBD-enriched preparations appear to show a more consistent, beneficial effect in ameliorating mood under different conditions, which could present some promise for future use in mood-related disorders. Most studies from cannabis abusers have shown that high concentrations of THC and high ratios of THC:CBD were associated with more vigorous euphoria, but also with higher anxiety, depression and psychotic symptoms. It has also been shown that CBD attenuates some effects of THC such as anxiety, cognitive deficits, and psychosis in heavy cannabis users, and high CBD:THC ratios produce an effect opposite to the high THC:CBD concentrations [11,145,175,178,222,223,224,225,226]. However, the mechanisms by which CBD and THC interact or alleviate negative consequences are not clear, due in part to the multiple biological targets that both compounds can modulate. There is an undeniable need to identify the roles of these targets of THC and CBD, including different receptor complexes to clarify the different, and in some cases, opposite biological effects on the brain. This is particularly important with the view of identifying novel therapeutic targets for multiple illnesses, including anxiety and depression, using cannabinoid preparations devoid of side effects apparent during chronic treatment, including after the use of very low dosages [227], as well as to possibly reverse the adverse consequences of long-term cannabis use and abuse or withdrawal signs in CUD individuals [11,145].

In summary, there are credible arguments for a functional role of the ECS in mood disorders including anxiety and depression. There is also immense interest regarding clinical research in cannabinoids, in part due to the fact that cannabis was declared to be an illicit drug for a long time, hindering its legitimate use in medicine if proven to have benefit. In mood disorders, there is a need for multiple comprehensive RCTs, as the data so far are not completely convincing.

## Figures and Tables

**Figure 1 brainsci-13-00325-f001:**
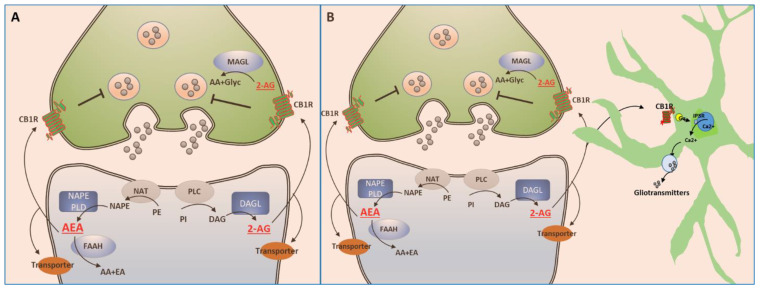
Schematic representation of the endocannabinoid signaling system. Endocannabinoids (eCBs) 2AG (2-arachidonoylglycerol) and AEA (anandamide) are synthesized in postsynaptic neurons and act pre-synaptically at their receptors (CB1Rs) in a retrograde mode (**A**). eCBs can also activate CB1R localized post-synaptically in some cases. In general, activation of CB1 receptors by eCBs decreases the probability of neurotransmitter release through multiple mechanisms, comprising inhibition of calcium influx and activation of potassium channels These effects are terminated by eCB reuptake followed by degradation: 2AG by MAGL pre-synaptically and AEA by FAAH post-synaptically. A large proportion (45–48%) of synapses in different brain regions is tripartite (**B**), formed by pre-synaptic and post-synaptic neurons, and a glial cell—an astrocyte—which expresses CB1R. In response to neuronal activation, eCBs released by the postsynaptic neurons can activate CB1R expressed on astrocytes. The astrocytes respond with an increase in intracellular calcium leading to release of gliotransmitters (**B**).

**Figure 2 brainsci-13-00325-f002:**
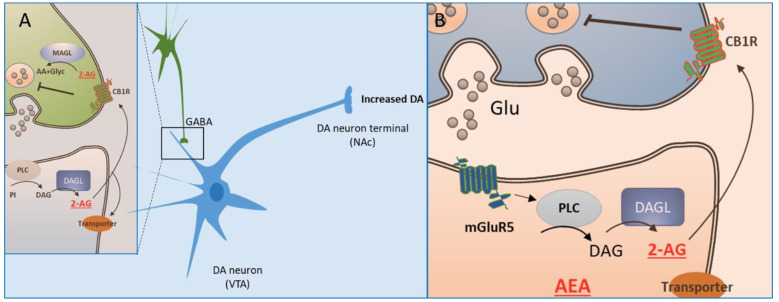
Modulation of neuronal activity by the endocannabinoid system. Endocannabinoids modulate neuronal activity as illustrated for dopamine (DA) neurons with cell bodies in the ventral tegmental area (VTA). CB1 receptors on GABAergic terminals (**A**) can facilitate dopaminergic activity through suppression of the inhibitory input onto the GABA receptors present on DA neurons, leading to an increase of DA release. Further, the activation of mGluR5 receptor by released glutamate (**B**) leads to the synthesis and release of 2-arachidonoylglycerol (2-AG), which then retrogradely activates CB1 receptors to inhibit additional glutamate release.

**Table 1 brainsci-13-00325-t001:** Examples of preclinical studies showing the effects of cannabinoids on depression-like and anxiety-like behaviors.

Species	Context and Cannabinoid Used	Summary of Results	References
Rodents (rat, mice)	CB1 antagonist SR141716 Depression- or anxiety-like behavior	Depressive-like phenotype	[80,81]
Rodents (rat, mice)	CB1 receptor antagonists, Rimonabant and AM251	anxiogenic effects	[80,81,82]
Rodents (rat, mice)	Rimonabant Depression- or anxiety-like behavior	No effect	[80,83,84,85]
Rodents (rat, mice)	Rimonabant Depression- or anxiety-like behavior	Anxiolytic effect	[80]
Rodents (rat, mice)	CB1 agonists HU210 or oleamide Depression- or anxiety-like behavior	Anxiolytic and/or antidepression-like behavior	[10,68]
Rodents (rat, mice)	CB1 agonist THC Depression- or anxiety-like behavior	Anxiolytic and/or antidepression-like behavior	[10,69,70,80]
Rodents (rat, mice)	Synthetic CB1 agonists WIN55,212-2; CP55,940 Depression- or anxiety-like behavior	Anxiolytic and/or antidepression-like behavior	[10,71,72,80]
Rodents (rat, mice)	Higher doses of CB1 agonists Depression- or anxiety-like behavior	Pro-depressive and anxiogenic	[69,75,76,77,78,80]
Rodents (rat, mice)	The endogenous AEA Depression- or anxiety-like behavior	Anxiolytic and/or antidepression-like behavior	[73,74,80]
C57BL/6 mice	CBD Chronic stress	Treatment precludes anxiety-like behavior	reviewed in [80]
C57BL/6 mice	WIN55,212-2 Stress-induced anxiety	Reversal of social stress-induced anxiety-like phenotype	reviewed in [80]
C57BL/6 mice	MAGL inhibitor Chronic stress	Enhanced adult neurogenesis involved in anxiolytic like effects	reviewed in [80]
CB1 KO mice	Role of CB1 receptor in social anxiety and memory	Acute CB1 antagonist AM251 increased anxiety-like behavior in CB1 KOs and WT	reviewed in [80]
CB1 KO mice	Role of CB1 receptor in anxiety	Anxiogenic phenotype only under high light conditions	reviewed in [80]
CB1 KO mice	Role of CB1 receptor in stress coping behaviors	Increased passive stress coping behavior	reviewed in [80]
CB1 KO mice	Behavioral characterization of CB1-KO mice	Increased anxiety-like behavior. Reversed by cannabinoid antagonist	reviewed in [80]
DAGL-α KO mice	Role of DAGL-α, and 2-AG in anxiety-like behavior	Enhanced anxiety, stress and fear responses in DAGL-α KO	reviewed in [80]

See [10,67,68,69,70,71,72,73,74,75,76,77,78,80,81,82,83,84,85] for more details.

**Table 2 brainsci-13-00325-t002:** Examples of cannabinoid-based therapies to treat anxiety and/or depression.

Cannabinoid-Based Drug	Dosage (Where Indicated)	Outcomes for Anxiety/Depression	Related Condition	Systematic Reviews (References)
Cannabis	NA	↓Anxiety	HIV	Huang et al., 2016 Ref. [5]
Cannabis	NA	↓Anxiety ↓Depression ↑Quality of life	Antidepressant, anxiolytic and quality of life	Black et al., 2019 Chadwick et al., 2020 Refs. [108,109]
*Cannabis sativa*	NA	↓Anxiety ↓Depression	Antidepressant	Black et al., 2019 Ref. [109]
Nabilone	NA	↓Anxiety	Fibromyalgia	Huang et al., 2016 Ref. [5]
Nabilone	NA	↓Anxiety and Stress	Cancer	Huang et al., 2016 Ref. [5]
Nabilone	28 days of 1 mg	↓Anxiety	Psychoneurotic anxiety disorder	Stanciu et al., 2021 Ref. [171]
Nabilone	once 2 mg, then weekly 0.5–5 mg for five weeks	No improvement in anxiety.	Anxiety neurosis or generalized anxiety disorder	Stanciu et al., 2021 Ref. [171]
Dronabinol [THC]	NA	↓Anxiety	Chronic neuropathic pain	Huang et al., 2016 Ref. [5]
Dronabinol (oral)	16.6 mg (7.5–25 mg)	↓Anxiety ↓Depression ↑Quality of life	Analgesic and impact on mental health outcomes	Black et al., 2019 Ref. [109]
Dronabinol (oral)	5–10 mg	↓Depression ↑Quality of life	Antidepressant and quality of life	Black et al., 2019 Chadwick et al., 2020 Refs. [108,109]
Dronabinol (oral)	10 mg (2.5–10 mg)	↓Anxiety ↑Quality of life	Antidepressant and quality of life	Black et al., 2019 Chadwick et al., 2020 Refs. [108,109]
THC	7 days of 0.3 mg/kg	No improvement in depressive symptoms.	Unipolar and bipolar major depression	Stanciu et al., 2021 Ref. [171]
CBD	4 weeks of 300 mg	↓Anxiety	Social anxiety disorder	Stanciu et al., 2021 Ref. [171]
CBD	Single dose of 600 mg	↓Anxiety	Social anxiety disorder	Stanciu et al., 2021 Ref. [171]
CBD	Single dose of 400 mg	↓Anxiety	Social anxiety disorder	Stanciu et al., 2021 Ref. [171]
Sativex [THC+CBD]	NA	↓Anxiety ↑Quality of life	Diabetic-neuropathy	Huang et al., 2016 Ref. [5]
CBD + THC	NA	Some but not all studies: ↓Anxiety ↓Depression ↓Stress ↑Quality of life	PTSD	Orsolini et al., 2019 Ref. [172]

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
