# Peer review of "Endocannabinoid System and Exogenous Cannabinoids in Depression and Anxiety: A Review"

_brainsci, 2023, doi:10.3390/brainsci13020325_

Round 1

Reviewer 1 Report

This is a very interesting paper exploring the relationship between the endocannabinoid system and exogenous cannabinoids in patients suffering from depressive disorders and anxiety. The paper is well-written and of interest for the readers; however, several changes are recommend to improve the paper.

Abstract.

1- The abstract is really short. I recommend to expand the rationale of the paper and results. Why is important to investigate the association between cannabinoids and depressive and anxiety disorders?

2-At the end of the abstract, the authors report that "evidence is presented for the clinical and use of them. I recommend to summarize the results and provide a conclusion in the abstract section.

Introduction.

1- The introduction is mainly focused on cannabis. I recommend to expand this section, also introducing that are many endogenous systems capable of influencing depressive and anxiety systems.

2-The third paragraph is mentioning the main aims and objectives of the paper. I recommend to expand this section by including the main and secondary objectives of the paper. A first step would be to review the endocannabinoid system, mainly the cannabinoid receptors, and the additional receptors. 

Methods.

1- The authors carried out a narrative review on endocannabinoids systems and the association with depression and anxiety. How did the authors the electronic searches? I recommend to add information about the screening and selection processes. What inclusion and exclusion criteria did they use?

Results

I recommend to add some tables and figures to illustrate findings. For instance, a possibility would be to summarize results from animals models and human studies in terms of influence of cannabinoids on depressive and anxiety symptoms.

Discussion, conclusions and future perspectives should be separated. I recommend to explain them into separate sections.

Author Response

Reviewer 1

This is a very interesting paper exploring the relationship between the endocannabinoid system and exogenous cannabinoids in patients suffering from depressive disorders and anxiety. The paper is well-written and of interest for the readers; however, several changes are recommend to improve the paper.

 We thank the reviewer for the positive comments. We implemented the reviewer’s suggestions as follows

Abstract.

1-      The abstract is really short. I recommend to expand the rationale of the paper and results. Why is important to investigate the association between cannabinoids and depressive and anxiety disorders?

The abstract has been expanded as the reviewer suggested.

2-At the end of the abstract, the authors report that "evidence is presented for the clinical and use of them. I recommend to summarize the results and provide a conclusion in the abstract section.

 The abstract has been expanded as the reviewer suggested and includes in the revised version a summary and a conclusion.

Introduction.

1-      The introduction is mainly focused on cannabis. I recommend to expand this section, also introducing that are many endogenous systems capable of influencing depressive and anxiety systems.

We have added the reviewer’s suggestion in lines 51-54. It reads: “It is well known that other endogenous systems such as dopamine, serotonin, opioid, adrenergic systems are also involved in depression and anxiety, but the main focus of the present review is on the endogenous and exogenous cannabinoids in relation to depression and anxiety”. 

2-The third paragraph is mentioning the main aims and objectives of the paper. I recommend to expand this section by including the main and secondary objectives of the paper. A first step would be to review the endocannabinoid system, mainly the cannabinoid receptors, and the additional receptors. 

This paragraph has been changed to reflect the reviewer’s suggestion. It reads as follows:

We describe the most important constituents of Cannabis sativa, review the endocannabinoid system, including the different cannabinoid receptors, and the machinery involved in the synthetize, transport and degradation of the endocannabinoids. This is followed by the most established evidence regarding depression/anxiety and ECS, before reviewing the relevant effects of different exogenous cannabinoid preparations in relation to depression and related ailments.

Methods.

1-      The authors carried out a narrative review on endocannabinoids systems and the association with depression and anxiety. How did the authors the electronic searches? I recommend to add information about the screening and selection processes. What inclusion and exclusion criteria did they use?

The present review is a general review on the thematic: depression and cannabinoids. This is not a systematic review where certain criteria are used for selection of papers to be included or excluded. Therefore, our search was general and included words around the thematic: [cannabinoids; depression; anxiety; clinical use or cannabinoids] among other MESH for our online research.

Results

I recommend to add some tables and figures to illustrate findings. For instance, a possibility would be to summarize results from animals models and human studies in terms of influence of cannabinoids on depressive and anxiety symptoms.

We added two tables: [Table 1: line 217] and [Table 2: line 421] as suggested by the reviewer.

Discussion, conclusions and future perspectives should be separated. I recommend to explain them into separate sections.

Unfortunately, the different parts [Discussion, conclusions and future perspectives] are imbricated due to the different and numerous lacunes in understanding the conflicting results, depending on the doses, the formulas, the illness associated (co-morbidity) with depression or anxiety {[PTSD; MDD; cancer, HIV, etc..], and a total lack of any guidance on the therapeutic use of cannabinoids for any of these different conditions. There is no simple conclusion and each part needed a single discussion and perspectives.

Reviewer 2 Report

Dear Editor,
I really appreciate the opportunity to review the manuscript brainsci-2177886 entitled:
"Endocannabinoid system and exogenous cannabinoids in depression and anxiety: a review"

I commend the authors for describing this critical and timely issue. The paper is interesting and well-written; however, I would like to highlight some issues that merit revision:

The review is really well done and very detailed; I may have missed but not how that now widespread self-treatment represented by "low" dosage cannabis derivatives, sold to virtually anyone legally in many countries, was analyzed. Are there implications in the pathologies in question? I ask the authors to at least mention this issue in the manuscript to supplement the submitted work

Author Response

Reviewer 2

Dear Editor,
I really appreciate the opportunity to review the manuscript brainsci-2177886 entitled:
"Endocannabinoid system and exogenous cannabinoids in depression and anxiety: a review"

I commend the authors for describing this critical and timely issue. The paper is interesting and well-written; however, I would like to highlight some issues that merit revision:

The review is really well done and very detailed; I may have missed but not how that now widespread self-treatment represented by "low" dosage cannabis derivatives, sold to virtually anyone legally in many countries, was analyzed. Are there implications in the pathologies in question? I ask the authors to at least mention this issue in the manuscript to supplement the submitted work

Answer: We thank the reviewer for the positive comments. In the systematic reviews that we included in our manuscript, different doses were used including low doses in different conditions associated with depression (co-morbidity). We added a summary in Table2. We also have now cited a review, which is focused specifically on very low doses [222: Sarne Y. (2019) Beneficial and deleterious effects of cannabinoids in the brain: the case of ultra-low dose of THC. The American Journal of Drugs and Alcohol abuse 45(6): 551-562].
